# Progressive Failure Analysis for 5MW-Class Wind Turbine Composite Blades with Debonding Damage based on CZM Method

Yunjung Jang [1], Hakgeun Kim [2] and Kiweon Kang [2],*

1    Institute of Offshore Wind Energy, Kunsan National University, Gunsan 54150, Republic of Korea
2    Department of Mechanical Engineering, Kunsan National University, Gunsan 54150, Republic of Korea
*    Correspondence: kwkang68@kunsan.ac.kr; Tel.: +82-63-469-4872

**Abstract:** Composite wind turbine blades may experience interlaminar damage, including adhesive failure, cracking, and interlaminar fracture failure, from manufacturing or external fatigue load. Among these, the adhesive failure of adhesive joints is critical. Therefore, it is important to identify the failure mechanism in the adhesive joints between the spar cap–shear web and trailing edge of composite blades. We calculated fracture toughness through Mode I, Mode II, and Mixed-mode tests for quantitative analysis of adhesive joints. Then, to select a modeling method for realizing the damage generated in the blade, the method was verified from the specimen level. A damage model was constructed, considering contact conditions of the spar cap–shear web and trailing edge for an NREL 5MW wind turbine blade. Finally, a damage model based on cohesive zone modeling was used to analyze the progressive failure behavior of debonding at adhesive joints according to the external force applied to the blade.

**Keywords:** adhesive joint damage; wind turbine composite blade; interlaminar fracture toughness; laminated biaxial/triaxial hybrid; mixed-mode bending

## 1. Introduction

While composite blades are designed to have a design life of over 25 years, various damages have been reported in real composite blades. Among them, it is reported that the debonding damage in the spar cap–shear web joint is the most frequently occurring and dangerous damage mode. To solve these problems, a methodology has been developed to explain the initiation of the debonding effect as a source of major damage in blades and its growth due to loading in the adhesive joint area, and the corresponding mechanism for this methodology has been verified [1].

Wind turbine blades are designed by laminating unidirectional, biaxial, and triaxial composite materials in a hybrid form to improve structural performance and reduce weight. This improves the load resistance of the upper and lower skins directly under the load and the shear webs supporting them. This hybrid form utilizes composite materials with different characteristics to offset the shortcomings of each lamination composite material; therefore, the upper and lower layers vary in properties [2,3]. The mechanical properties evaluated for each composite material are considered in this type of design; however, the failure characteristics for the hybrid composite materials may differ from the performance of a single material [4]. In the existing research trend on the initiation and growth of debonding damage, Eder et al. [5] conducted specimen-level tests, crack detection, and theoretical analyses for the adhesive joint of the trailing edge (TE). Othman et al. [6] and Robert et al. [7] proposed methods for detecting and testing the interlaminar debonding and delamination damage of wind turbine blades. Hasan et al. [8] conducted a debonding analysis on the T-joints of wind turbine blades, and Ji et al. [9] developed a fracture mechanics approach for the failure of the joints of wind turbine blades. In this research, the

test data for Mode II, that is, the sliding mode, were acquired, and the CZM method of the spar–web joint area was used to propose a methodology for the initiation and growth of the debonding damage in wind turbine blades. However, in addition to the spar cap–shear web area, the adhesive joint of blades encompasses the TE and leading edge (LE) joints where the suction and pressure sides are bonded. This TE/LE adhesive joint is treated as the most vulnerable structural part of wind turbine blades [10–13]. Therefore, to study the initiation and growth of debonding damage to blade joints, it is crucial to acquire test data for Mode I (DCB), Mode II (ENF), and Mixed-mode (MMB) and verify the material properties.

As debonding damage in the adhesive joint of blades initiates and grows with the externally applied load, the forces acting on the wind turbine must be reviewed. Particularly, if the wind blade is operated under various environmental conditions, the load applied to the blade leads to combined load conditions. These combined loads have a significant adverse effect on the structural integrity of blades compared to the simple load condition in which only a unidirectional load is applied [14]. Therefore, the debonding damage in the adhesive joints of wind blades should be analyzed under combined load conditions.

In this study, a model was proposed for predicting the initiation/growth of debonding damage in the adhesive joint area of wind turbine composite blades. To this end, a blade model using the CZM method was constructed for the adhesive joint area, including the TE joint, which is the most vulnerable part of the NREL 5MW-class composite blade. In addition, Mixed-mode tests and analyses were performed in accordance with ASTM D6671 [15] based on the fracture toughness calculated through the ASTM D5528 (Mode I) test [16] and D7905 (Mode II) tests [17,18] in a previous study [19]. Moreover, the combined load calculated in the integrated load analysis [14] was applied to the blade model to simulate the effect of the combined load borne by the blade. Through the above process, a prediction model was proposed for the initiation/growth of debonding damage in adhesive joints. Based on this model, the initiation/growth mechanisms of debonding damage were analyzed.

## 2. Experimental and Numerical Failure Analysis of Adhesive Joints

### 2.1. Theoretical Background

Because the load conditions applied to a composite blade involve combined loads, the interaction between the opening, shearing, and Mixed-mode should be considered. To consider this interaction, the analytical method for modeling an adhesive was performed in two steps in this study. First, as a step for evaluating initial failure based on the strength and rigidity of the adhesive, the quadratic normal stress criterion proposed by Gui et al. [20] was applied, as shown in Equation (1). Second, because crack growth occurs after the initial failure, a progressive failure analysis was performed by applying the criterion proposed by Benzeggagh and Kenane [21], as shown in Equation (2), which is an energy release rate-based crack growth criterion that reflects the Mixed-mode condition.

$$\left(\frac{\sigma_n}{N_{max}}\right)^2 + \left(\frac{\sigma_s}{T_{max}}\right)^2 + \left(\frac{\sigma_t}{S_{max}}\right)^2 = 1 \tag{1}$$

$$G_{IC} + (G_{IIC} - G_{IC})\left(\frac{G_{II}}{G_I + G_{II}}\right)^\eta = G_c \tag{2}$$

where $N_{max}$ is the interlaminar tensile strength, $T_{max}$ is the interlaminar shear strength, $S_{max}$ is the transverse interlaminar shear strength, $\sigma_n$ is the nominal stress, $\sigma_s$ is the shear stress, $\sigma_t$ is the transverse shear stress, $G_{IC}$ is the critical fracture toughness of Mode I, $G_{IIC}$ is the critical fracture toughness of Mode II, $G_I$ is the fracture toughness of mode I, $G_{II}$ is the fracture toughness of mode II, $G_C$ is the critical fracture toughness of Mixed-mode, and $\eta$ is the Mixed-mode material factor.

### 2.2. Specimen Preparation

Test specimens were designed and prepared, as shown in Figure 1, for the fracture test; to determine the interlaminar fracture toughness of the biaxial/triaxial laminated hybrid composite. The specimens for the test of each failure mode were named as follows: (a) Mode I: double cantilever beam (DCB), (b) Mode II: end-notched flexure (ENF), and (c) Mixed-mode: mixed-mode bending (MMB). The specimens had dimensions of $175 \times 25$ mm and were laminated with three plies of biaxial glass fiber reinforced composite material at 1.68 mm (0.56 mm/ply) and two plies of triaxial composite material at 1.82 mm (0.91 mm/ply). A Teflon film with a thickness of 0.2 mm was inserted between the biaxial and triaxial laminations to implement the initial crack, and the remaining parts were attached using a cohesive. The crack lengths were 50 mm, 30 mm, and 25 mm for the DCB, ENF, and MMB, respectively. In addition, an aluminum hinge fabricated in-house was attached to connect the specimens to the load cell of the testing system during the test.

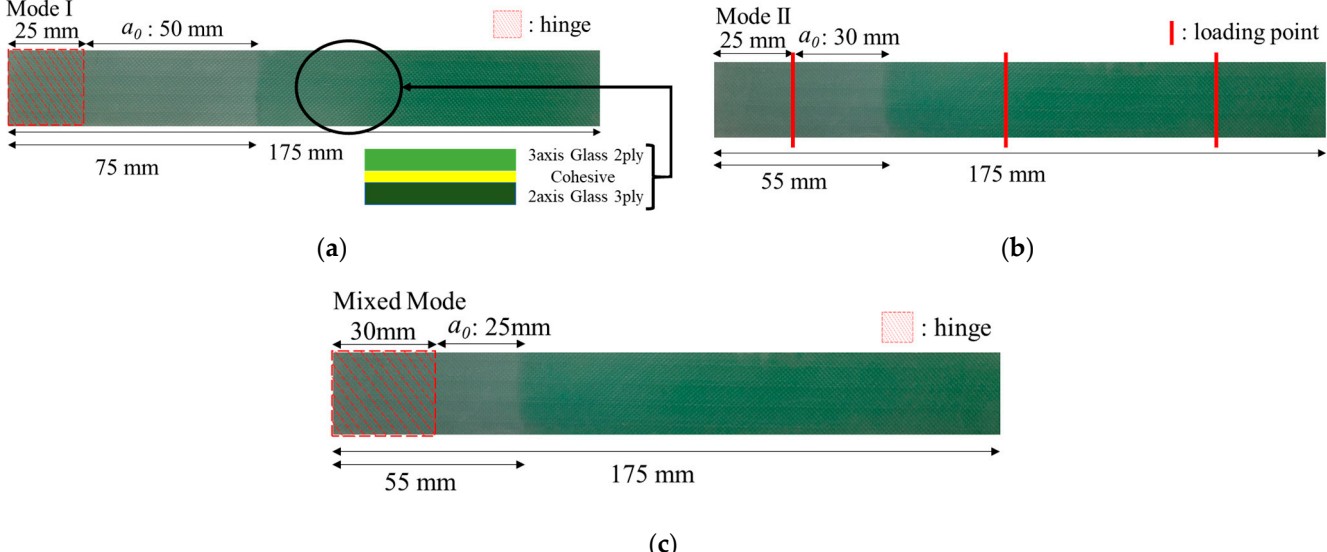

**Figure 1.** Test specimen dimension: (**a**) DCB (Mode I), (**b**) ENF (Mode II), and (**c**) MMB (Mixed-mode).

### 2.3. Experimental Test Procedure

For the interlaminar fracture test of the biaxial/triaxial hybrid composite, Mode I, Mode II, and Mixed-mode tests were performed using an Instron 8516 fatigue testing system (10 ton) equipped with a 5 kN load cell (Tovey Engineering. Inc., Phoenix, AZ, USA).

In the Mode I test, the lower part of the test specimen was fixed to the testing system using an aluminum hinge, and the aluminum hinge for the upper part was connected to the load cell to control the displacement at a speed of 1 mm/min, as shown in Figure 2a. The test was performed until the final length of the crack growth reached 50 mm, and a total of five tests were performed. Prior to proceeding with the interlaminar failure test, a preliminary test was added, in which the specimen was unloaded at a similar speed after the length of the crack reached 5 mm. The purpose of this test condition was to minimize the crack in the inhomogeneous adhesive joint during specimen preparation and improve the accuracy of the fracture toughness property.

In the Mode II test, the displacement was controlled at a speed of 0.5 mm/min in a three-point bending test form, in which both ends of the lower part of the test specimen were fixed while the middle of the upper part was pressed, as shown in Figure 2b. The test was performed until the final length of the crack was 10 mm and was repeated five times. Additionally, to eliminate the effect of the inhomogeneity in the adhesive joint on the fracture toughness property, the load was applied at a speed of 0.2 mm/min until crack growth occurred within 5 mm, and then the test was performed after unloading.

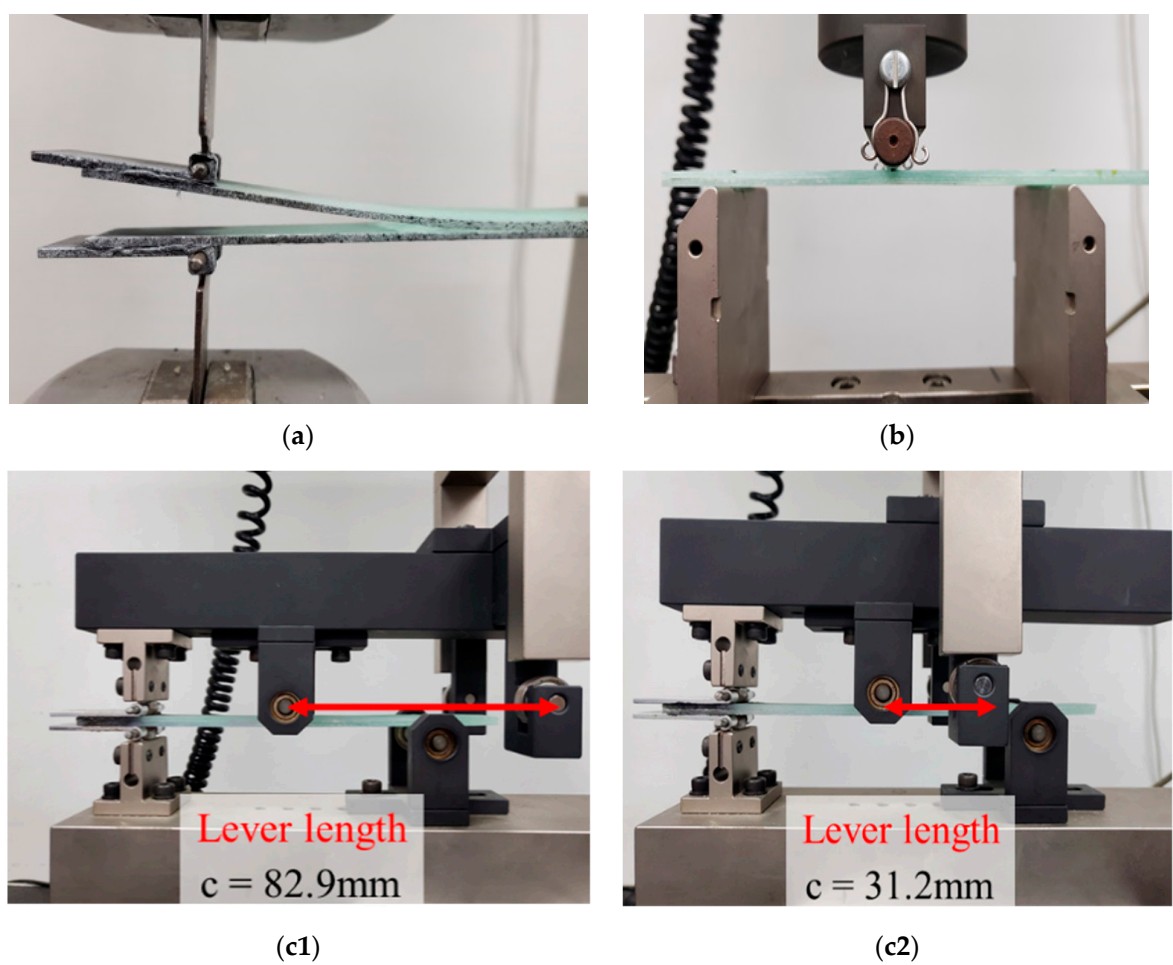

**Figure 2.** Fracture toughness test case: (**a**) Mode I, (**b**) Mode II, and (**c**) Mixed-mode.

The Mixed-mode test combines Mode I and Mode II to acquire the material factor that indicates the nonlinear behavior of crack growth [22,23]. To choose the lever length, c, which is the main condition for controlling the mixed-load mode, lever lengths of 82.9 mm and 31.2 mm were chosen for the test based on the mode mixture ratio, as shown in Figure 2c. In addition, the displacement was controlled at a speed of 0.8 mm/min until the final length of the crack growth reached 8 mm, and three tests were conducted for each lever length. At this time, to eliminate inhomogeneity in the adhesive joint, the test was performed at a speed of 0.5 mm/min until the crack growth reached 5 mm before unloading the specimen. Through these tests, the data for displacement, load, and crack growth length were obtained. The displacement and load data were obtained at a sampling rate of 10 Hz. In addition, the length of the crack growth was measured using a crack measurement camera, as shown in Figure 3, to acquire precise values, and the crack length was calculated using the Mercury RT software [24].

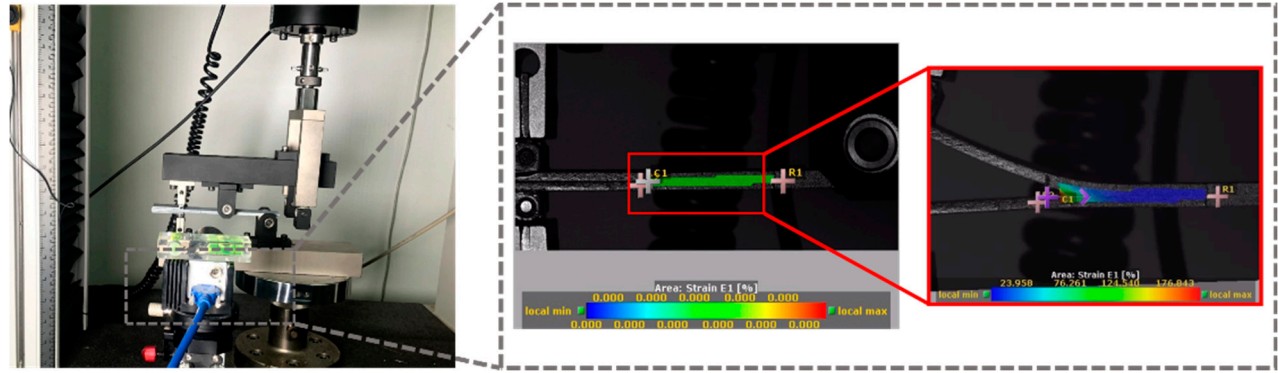

**Figure 3.** Crack measurement for fracture toughness of Mixed-mode.

## 2.4. FE Analysis—Numerical Modeling Method for Test Specimen-Level

To verify the crack growth model, a progressive failure analysis based on the CZM technique was performed using the ABAQUS [25] commercial software. S4R was used as the finite element model, and the COH3D8 element type was used as the adhesive. As shown in Figure 4, the element size of the adhesive was set to 0.1 mm to further improve the variability of the crack growth behavior and the convergence of the analysis. Tables 1 and 2 list the material properties of the adhesive used for preparing the test specimens and the glass fiber reinforced material, respectively, based on the material sheets provided by Human Composites Co., Ltd. (Gunsan-si, Republic of Korea) [26]. A load was applied up to 20 mm from the lever position through displacement control. The corresponding load, displacement, and crack growth length were measured to calculate the fracture toughness, and a comparative analysis of the test results was performed.

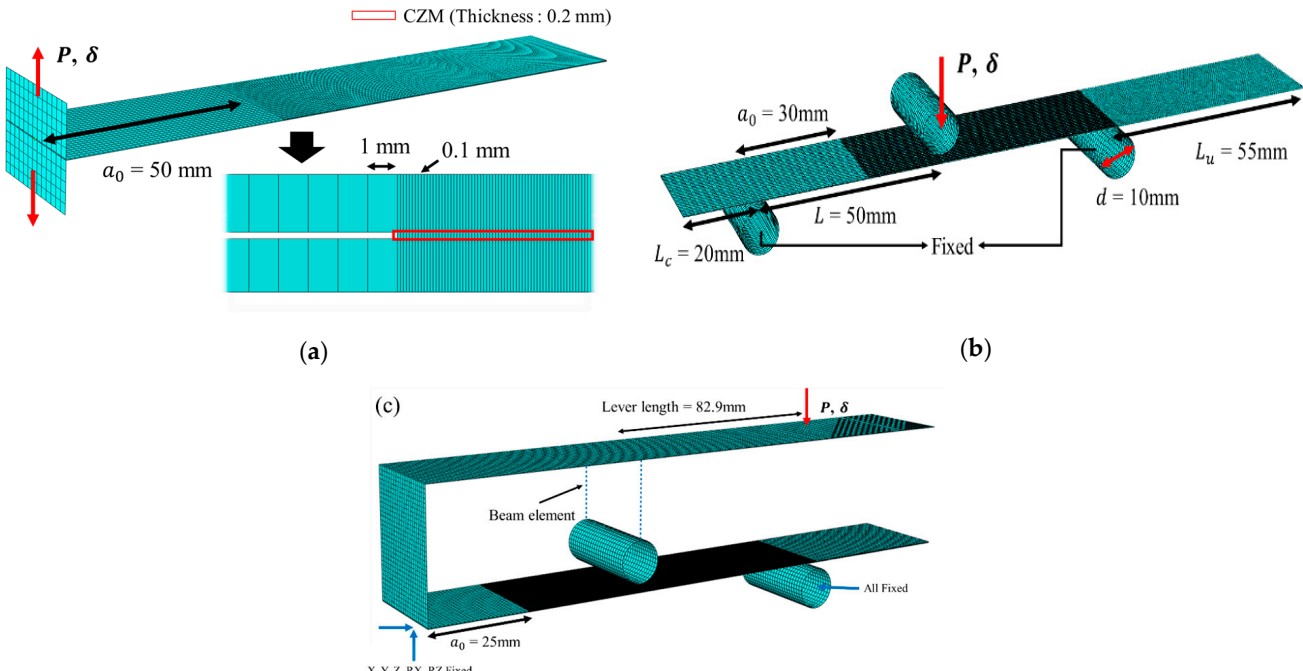

**Figure 4.** Finite element model of specimen: (**a**) Mode I, (**b**) Mode II, and (**c**) Mixed-mode.

**Table 1.** Material properties of FM73.

| | | |
|---|---|---|
| **Stiffness [MPa]** | $K_I$ | 4500 |
| | $K_{II}$ | 4270 |
| | $K_{III}$ | 4270 |
| **Strength [MPa]** | $S_I$ | 64.92 |
| | $S_{II}$ | 113 |
| | $S_{III}$ | 113 |

**Table 2.** Material properties of GFRP composites.

| | **3axis GFRP** | **2axis GFRP** |
|---|---|---|
| $E_{11}$ **[MPa]** | 26,700 | 10,900 |
| $E_{22}$ **[MPa]** | 13,300 | 10,900 |
| $G_{13}$ **[MPa]** | 74,600 | 11,600 |
| $\nu$ | 0.513 | 0.646 |
| $\rho$ **[kg/m$^3$]** | 2267 | 2243 |

*2.5. Composite Blade Numerical Modeling Method*

Damage modeling was performed using the CZM method at the test specimen level to analyze the debonding of the adhesive joint as the major source of damage in wind blades. The adhesive thickness of the blade joint was 10 mm, and a cohesive model was built for the spar cap–shear web and TE areas, as shown in Figure 5. For realized bonding with the blade model, the tie condition provided in the ABAQUS program was applied to fix the adhesive model, and the friction condition was applied to prevent interference between the structures. Based on this model, six degrees of freedom were constrained, namely, the three translational and the three rotational directions, as shown in Figure 6. For the load condition, the design load cases were calculated based on the specifications in Table 3, and for the ultimate load cases as listed in Table 4. Among these ultimate load conditions, a load equivalent to 200% of the $Mz_{min}$ load condition (DLC1.2f3) with a significant effect on adhesive failure was applied to analyze the characteristics of progressive failure after the initial failure occurred. In addition, for the analysis convergence method, a quasi-static analysis was performed to reflect the nonlinear convergence rate because the energy dissipation rate at the failure of the adhesive joint was significant. The analysis results were evaluated with a focus on the initial failure and the location of the failure based on the stress generated in the adhesive model relative to the load. The principal stresses generated in the blade and the maximum displacement of the blade were analyzed.

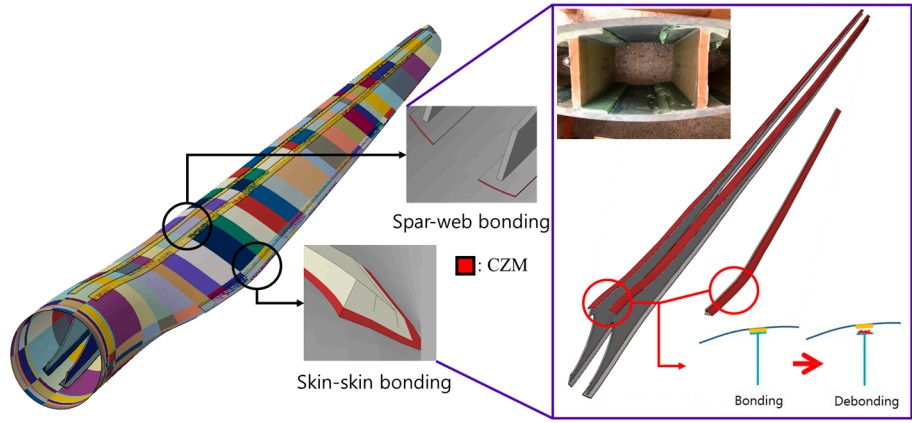

**Figure 5.** Schematic of the research objectives.

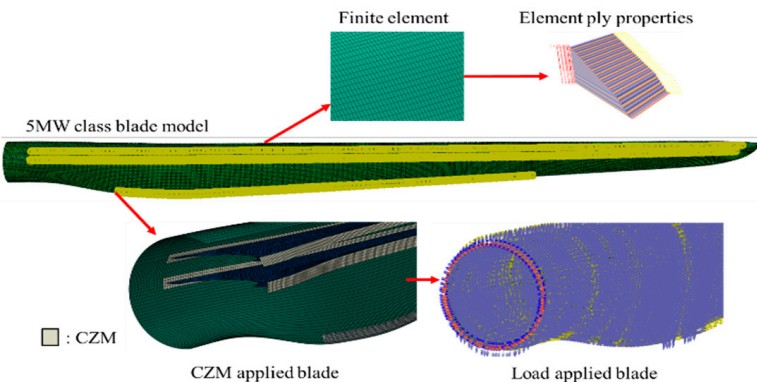

**Figure 6.** Boundary and load conditions in finite element analysis.

**Table 3.** Specifications for a 5 MW wind turbine [14].

| | | | |
|---|---|---|---|
| **Rated power (MW)** | 5 | **Blade set angle (°)** | 0 |
| **Class** | IIA | **Rotor shaft tilt angle (°)** | 5 |
| **No. of blades** | 3 | **Maximum chord length (m)** | 4.1 |
| **Blade length (m)** | 61.5 | **Rotor overhang (m)** | 5 |
| **Hub height (m)** | 90.55 | **Rotor position** | Upwind |
| **Tower height (m)** | 88.15 | **Transmission** | Gearbox |
| **Cut-in wind speed (m/s)** | 3 | **Power control** | Pitch |
| **Rated wind speed (m/s)** | 11.4 | **Fixed/Variable** | Variable |
| **Cut-out wind speed (m/s)** | 25 | **Gear Ratio** | 97 |
| **Rated rotational speed (rpm)** | 12.1 | **Substructure type** | Jacket |

**Table 4.** Ultimate loads at the blade root [14].

| | **Load Case** | | **Fx [kN]** | **Fy [kN]** | **Fz [kN]** | **Mx [kNm]** | **My [kNm]** | **Mz [kNm]** |
|---|---|---|---|---|---|---|---|---|
| **Fx** | **Max** | **dlc1.2k4** | 542.2 | −154.0 | 857.0 | 13,664 | 2784.7 | −385.1 |
| | **Min** | **dlc6.4b1** | −345.5 | −51.5 | −125.8 | −7659.2 | 1339.5 | 127.3 |
| **Fy** | **Max** | **dlc1.2k5** | 138.0 | 346.1 | 894.8 | 4482.3 | −8479.7 | −200.6 |
| | **Min** | **dlc1.2k5** | 193.6 | −270.5 | 826.3 | 1962.2 | 7784.2 | −147.6 |
| **Fz** | **Max** | **dlc1.2k5** | 112.6 | 64.5 | 1207.9 | 1622.7 | −1579.8 | −193.3 |
| | **Min** | **dlc6.4b3** | −137.7 | −6.10 | −236.5 | −4069.6 | 684.0 | 80.0 |
| **Mx** | **Max** | **dlc1.2f4** | 499.9 | −46.6 | 614.9 | 18,051 | 635.5 | −414.9 |
| | **Min** | **dlc6.4b3** | −297.5 | −59.2 | −203.7 | −8005.4 | 2009.3 | 151.5 |
| **My** | **Max** | **dlc1.2k5** | 193.6 | −270.5 | 826.3 | 1962.2 | 7784.2 | −147.6 |
| | **Min** | **dlc1.2k5** | 138.0 | 346.1 | 894.8 | 4482.3 | −8479.7 | −200.6 |
| **Mz** | **Max** | **dlc6.4b5** | −211.7 | −39.5 | −233.5 | −6822.9 | 2050.6 | 193.1 |
| | **Min** | **dlc1.2f3** | 535.6 | −204.3 | 838.8 | 17440 | 4034.7 | −472.5 |

## 3. Results

### 3.1. Fracture Toughness for Mode I, Mode II, and Mixed-Mode Tests

Through the Mode I test, the load value was obtained according to the displacement at the loading point, as shown in Figure 7. For the overall crack growth behavior encompassing this load and the critical crack point, the fracture toughness was calculated based

on Equation (3). Particularly in the crack length range of 50–53 mm, because the test is in the pre-crack (preliminary crack) state of producing a uniform initial crack tip, nonlinearity in the fracture toughness occurs with the crack growth length as the load increases.

$$G_{IC} = \frac{3P\delta}{2ba} \qquad (3)$$

where $P$ is the load, $\delta$ is the displacement, $b$ is the width of specimens, $a$ is the crack length, and $G_{IC}$ is the critical fracture toughness of Mode I.

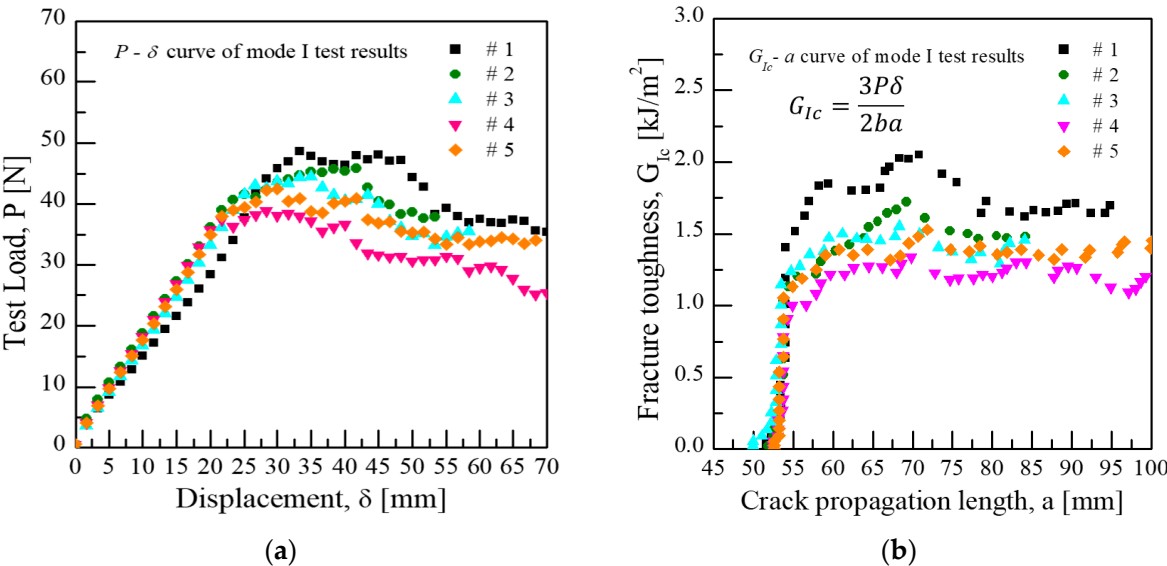

**Figure 7.** Fracture toughness test results of Mode I: (**a**) $P - \delta$ curve and (**b**) $G - a$ curve.

The results of the Mode II test based on ASTM D7905 are shown in Figure 8, and it was confirmed that the cracks grew from a load of 400 N on average. The fracture toughness for Mode II was obtained based on Equation (4).

$$G_{IIC} = \frac{9a^2 P\delta}{2b(2L^3 + 3a^3)} \qquad (4)$$

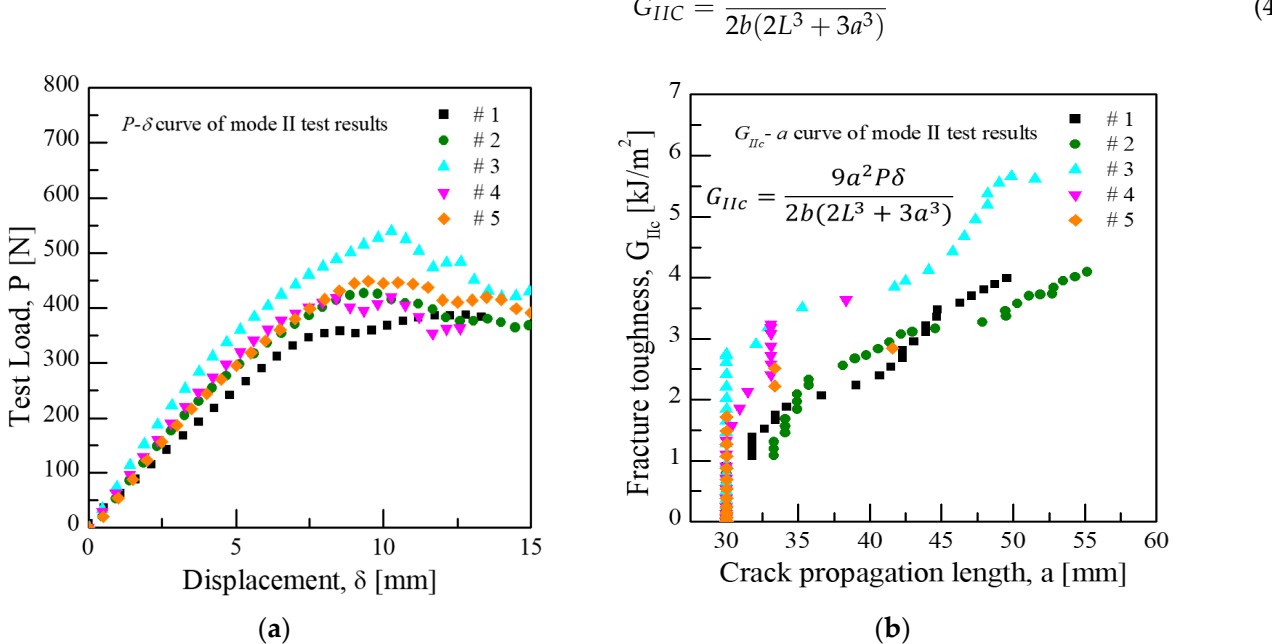

**Figure 8.** Fracture toughness test results of Mode II: (**a**) $P - \delta$ curve and (**b**) $G - a$ curve.

In addition, the deviation of the fracture toughness in Mode II was confirmed to be greater than that of Mode I. This phenomenon is attributed to the fiber bridging effect, which causes a high resistance to crack growth on the surface of the specimen. Furthermore, after the initial crack growth, the load and fracture toughness continued to increase at the crack length of 45 mm; this is believed to be the reason for the drastically slower crack growth rate as the crack growth point passed through the loading point, which reduced the shear directional load relative to the applied load.

The Mixed-mode test and fracture toughness calculation were performed in accordance with ASTM D6671. To select the location of the load for the Mixed-mode calculation, the non-dimensional crack length correction material constant was calculated using Equations (5)–(7) based on the elastic modulus of the material. The mode mixture transformation constant was calculated using Equations (8)–(9). Equation (8) is used to select the mode mixture ratio condition for the calculation of the material constant, $\eta$, of the B–K criterion. The level length, c, for the loading location was determined from Equation (10) based on the material and transformation constants for the parameters calculated through Equations (5)–(9).

$$\Gamma = 1.18 \frac{\sqrt{E_{11} E_{22}}}{G_{13}} \tag{5}$$

$$\chi = \sqrt{\frac{E_{11}}{11 G_{13}} \left\{ 3 - 2 \left( \frac{\Gamma}{1+\Gamma} \right)^2 \right\}} \tag{6}$$

$$\beta = \frac{\alpha + \chi h}{\alpha + 0.42 \chi h} \tag{7}$$

$$G_T = G_I + G_{II} \tag{8}$$

$$\alpha = \frac{1 - \frac{G_{II}}{G_T}}{\frac{G_{II}}{G_T}} \tag{9}$$

$$c = \frac{12 \beta^2 + 3\alpha + 8\beta \sqrt{3\alpha}}{36 \beta^2 - 3\alpha} \tag{10}$$

where $\Gamma$ is the transverse modulus correction parameter, $\chi$ is the crack length correction parameter, $\alpha$ is the mode mixture transformation parameter for setting the lever length, and $\beta$ is the non-dimensional crack length correction for the mode mixture.

The mode mixture ratios ($G_{II}/G_T$) were set to 0.23 and 0.64, and the Mixed-mode test was conducted at the corresponding lever lengths of 82.9 mm and 31.2 mm, respectively. The fracture toughness was calculated using Equations (11)–(13). Equations (11) and (12) are used for calculating the fracture toughness with characteristics of Mode I and II in the Mixed-mode, respectively. The critical fracture toughness for the Mixed-mode was calculated using Equation (13), which combines the criteria formula for Mode I and II.

$$G_{(I)C} = \frac{12 P^2 (3c - L)^2}{16 b^2 h^3 L^2 E_{1f}} (\alpha + \chi h)^2 \tag{11}$$

$$G_{(II)C} = \frac{9 P^2 (c - L)^2}{16 b^2 h^3 L^2 E_{1f}} (\alpha + 0.42 \chi h)^2 \tag{12}$$

$$G_{(I+II)C} = G_{(I)C} + G_{(II)C} \tag{13}$$

To calculate the critical fracture toughness at the mode mixture ratio, load and displacement data were obtained, as shown in Figure 9a. The critical point is shown in Figure 9b, and it was selected as deviation from linearity (NL), which analyzes fracture toughness through a criterion that deviates from linearity based on load and displacement. Table 5 lists the fracture toughness at the critical points for each mode mixture ratio. That is, the NL critical fracture toughness was found to be 1.07 kJ/m$^2$ and 1.82 kJ/m$^2$ at the mode mixture

ratios of 0.23 and 0.64 and under loads of 107.2 N and 340.2 N, respectively. A material constant of 1.85 was calculated by comparing the $G_{(I+II)C}$ value from the Mixed-mode test with the $G_c$ value based on the B–K criterion. Figure 10 shows this result as a graph, which demonstrates that the Mode I characteristics are more distinct as the lever length increases, whereas the Mode II characteristics are dominant when the lever length decreases.

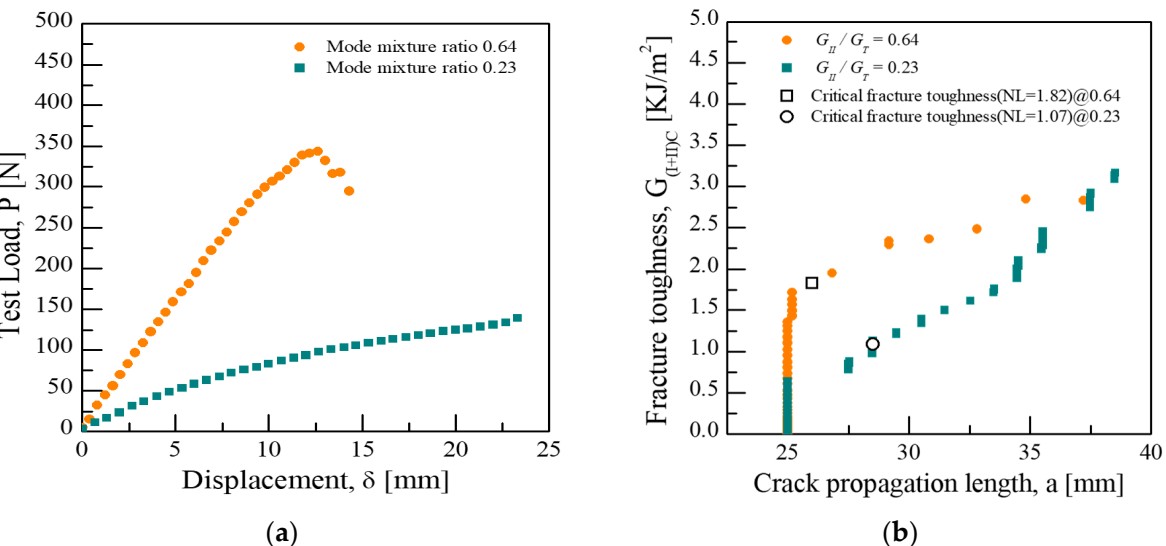

| (a) | (b) |

**Figure 9.** Mixed-mode fracture toughness test results: (**a**) $P - \delta$ curve and (**b**) $G - a$ curve.

**Table 5.** Experimental result of Mixed-mode.

| Mode Mixture Ratio ($G_{IC}/G_{IIC}$) | 23% | 64% |
|---|---|---|
| $c$ [mm] | 82.9 | 31.2 |
| $P$ [N] | 107.2 | 340.2 |
| $G_{(I+II)C}$ [kJ/m²] | 1.07 | 1.82 |

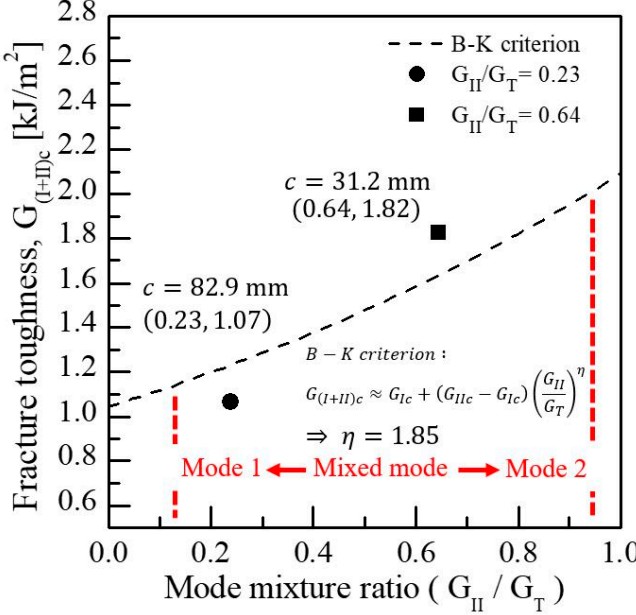

**Figure 10.** B–K criterion curve of Mixed-mode test results.

On the one hand, to examine the fracture surface of the specimens following the test, the fracture surface was observed at each mode mixture ratio, as shown in Figure 11, and the crack growth behavior and fracture shapes were confirmed to vary with the mode mixture ratio. In the form of the fracture interface, fall off a fiber and failure proceeded with bridging on the adhesive surface of the upper side, and an adhesive layer was confirmed on the lower side.

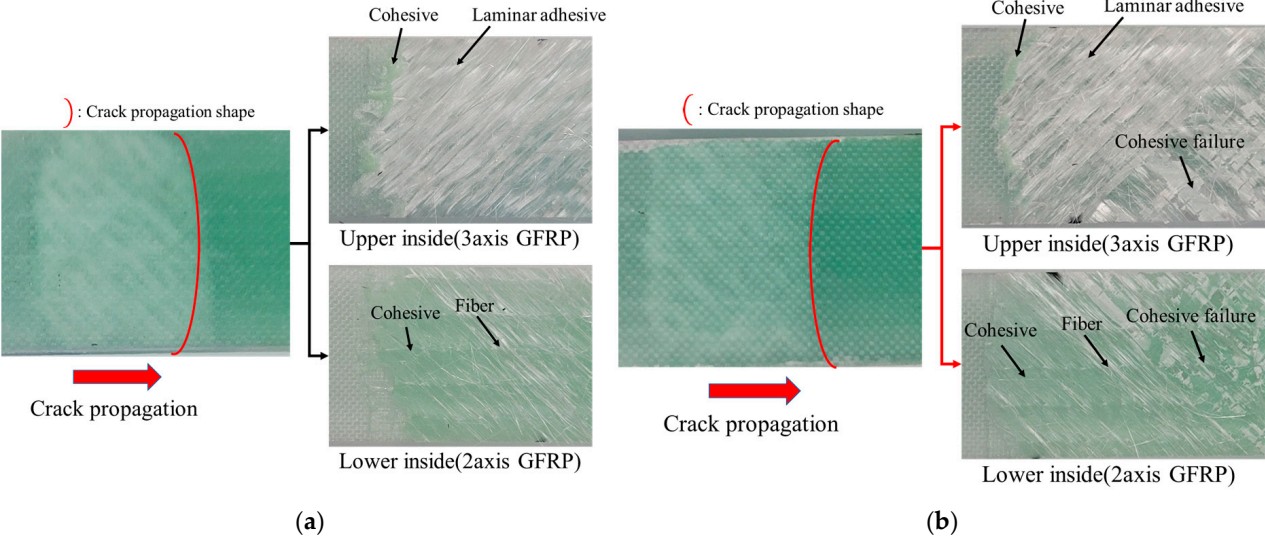

**Figure 11.** Fracture cross-section of Mixed-mode test results: (**a**) c = 82.9 mm, mode mixture ratio = 0.23; and (**b**) c = 31.2 mm, mode mixture ratio = 0.64.

### 3.2. FE Analysis for Mode I, Mode II, and Mixed-Mode Tests

To analytically evaluate the crack growth behavior according to each test mode, the fracture toughness was calculated under the same test conditions, as shown in Table 6. Through NL analysis for each test mode, we selected values of 1.12 kJ/m$^2$ and 2.02 kJ/m$^2$ for Modes I and II, respectively. For Mode III, we used the same value as Mode II [27]. In addition, for the Mixed-mode, we selected a material factor of 1.85. First, in Figure 12a,b, which show the displacement–load curve as the analysis result under Mode I conditions and the crack growth behavior and fracture toughness, respectively, it can be observed that the test and analysis exhibited similar trends. A pre-crack test was performed at 3 mm to ensure the homogeneity of the initial crack shape. Additionally, the analysis considered that crack growth behavior was confirmed from 50 mm (including pre-crack length) because crack propagation was generally simulated as a cohesive failure.

**Table 6.** Fracture toughness test results of hybrid composites.

|  |  | 2axis$_3$/FM73/3axis$_2$ |
|---|---|---|
| **Fracture toughness [kJ/m$^2$]** | **Mode I, $G_I$** | 1.12 |
|  | **Mode II, $G_I$** | 2.02 |
|  | **Mode III, $G_{III}$** | 2.02 |
| **Material constant** | **B–K criterion [Mixed-mode], η** | 1.85 |

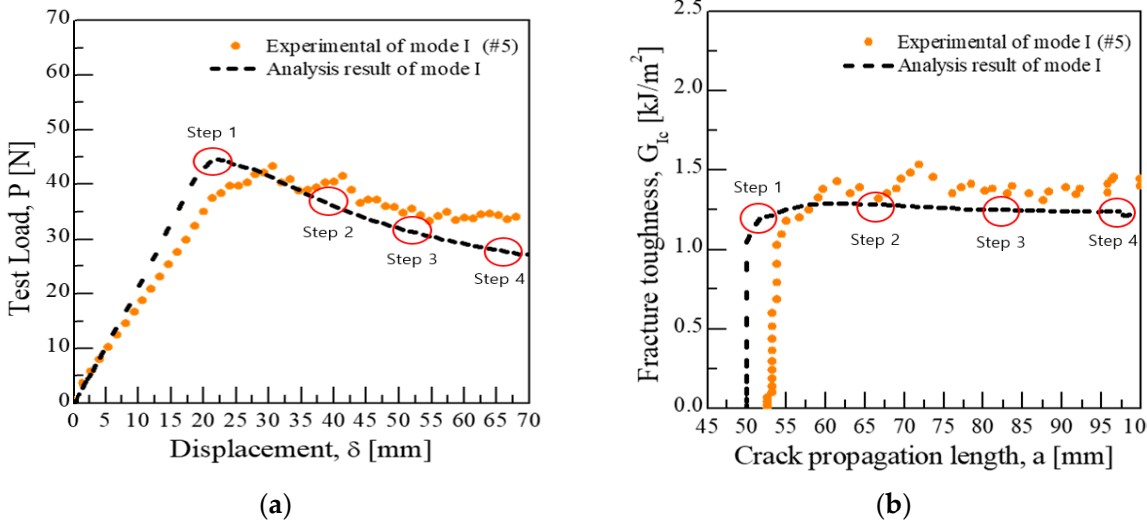

**Figure 12.** Comparative analysis of specimen-level FE analysis results and Mode I test results: (**a**) $P - \delta$ curve and (**b**) $G_I - a$ curve.

From Figure 13, it can be deduced that the crack grew as the stress propagated from the center of the specimen to the edge. Figure 13a shows the initial crack selection, the shape of the final crack, and the in-plane stress as the principal stress. Figure 13b shows the crack growth behavior with the applied displacement and the average stress at the critical point. Particularly, Figure 13b shows a similar trend as the fracture surface shown in Figure 11a based on the locations of the initial crack and crack growth. In other words, the reliability of the crack growth model in Mode I was established by confirming the similarity between the progressive fracture analysis method implemented using finite element analyses and the test.

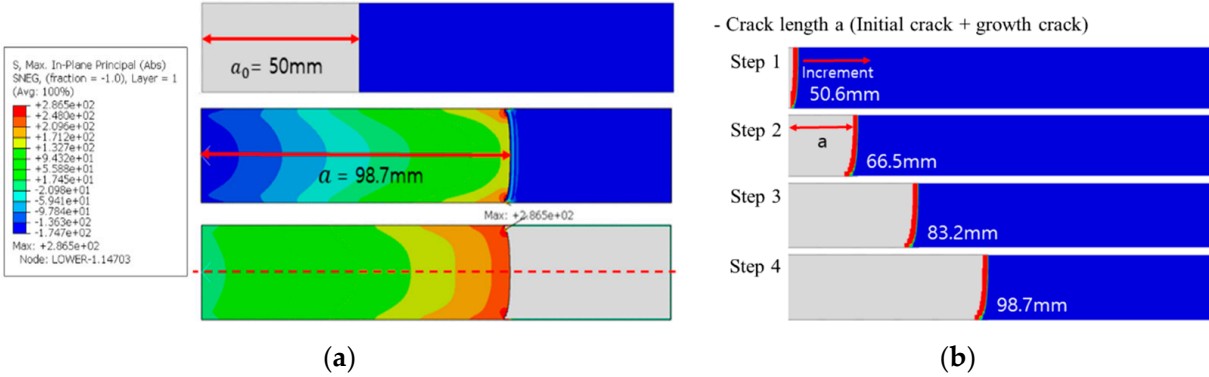

**Figure 13.** FE analysis result shape of Mode I: (**a**) in-plane stress contour and (**b**) crack growth length at applied displacement.

To analytically evaluate the crack growth behavior of Mode II, a finite element analysis was performed under the same test conditions. Figure 14a shows the resulting displacement–load curve, which indicates that the deviation between the test and analysis result varied from that of Mode I. It was found that the softening effect led to greater non-linearity in the analysis compared to that in the test [28]. In addition, as seen in Figure 14b, which illustrates the fracture toughness with respect to the crack growth length, similar trends were observed at the location of the initial crack, whereas an error began to manifest during the crack growth due to the softening effect. Furthermore, it is believed that the rapid change in fracture toughness at a crack growth length of 45 mm was due to the

increase in the load relative to the crack growth rate as the location of the crack growth approached the loading point.

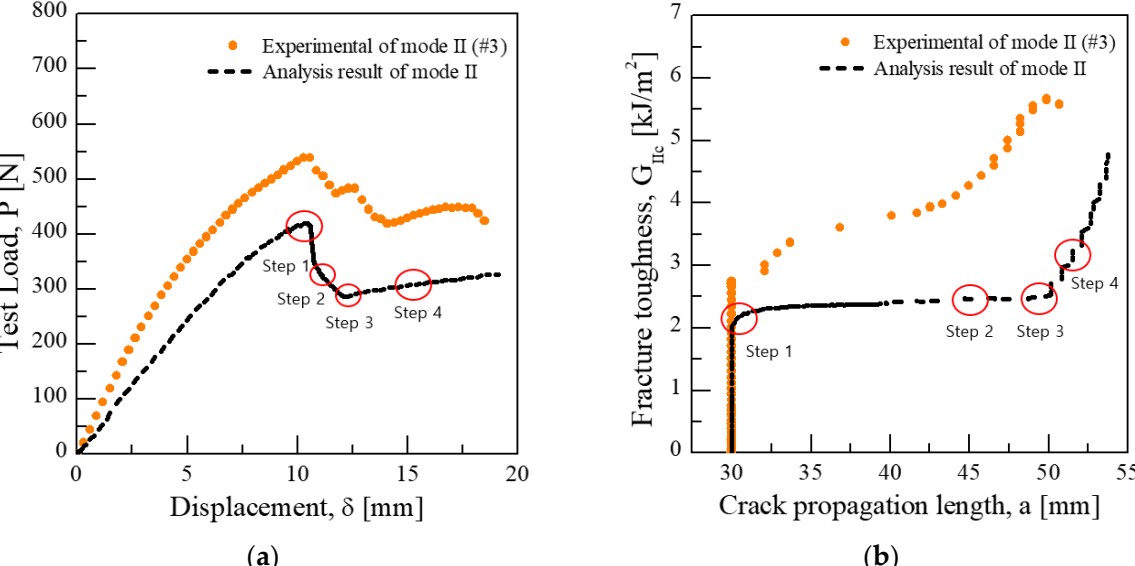

**Figure 14.** Comparative analysis of specimen-level FE analysis results and Mode II test results: analysis result shape of Mode I: (**a**) $P - \delta$ curve and (**b**) $G_{II} - a$ curve.

To closely analyze this crack behavior, the crack growth length and stress propagation with respect to the load-displacement were examined, as shown in Figure 15. Figure 15a shows the final crack growth length and the principal stress shape in the composite material. In contrast to Mode I, the crack appears to propagate from the edge to the center of the specimen. This is similar to the fracture surface based on the locations of the initial crack initiation and crack growth shown in Figure 11b. Figure 15b illustrates the shape of the average stress on critical point with respect to crack growth, which indicates that the crack propagated nonlinearly in the center as well as on the edge at the length of 45 mm and beyond. This phenomenon was observed in the crack growth located near the loading point, and similar crack growth behaviors were confirmed in the test and the analysis.

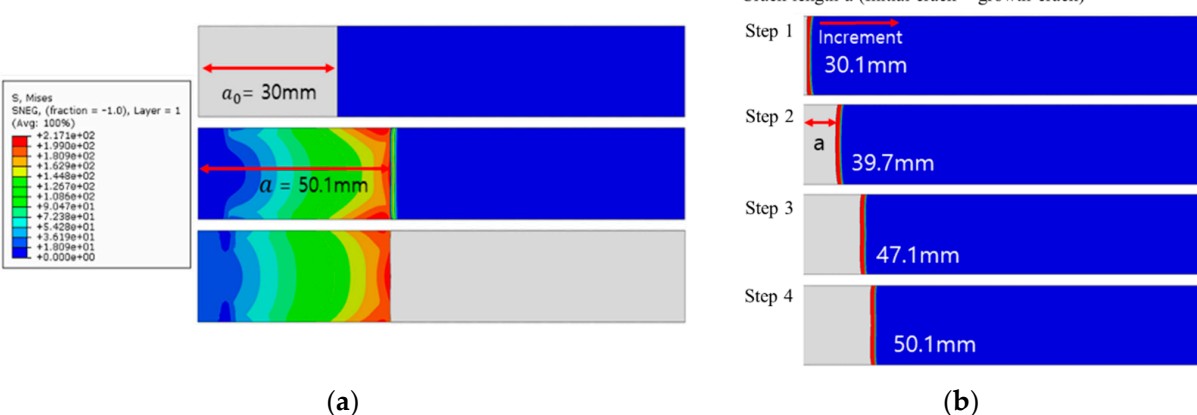

**Figure 15.** FE analysis result shape of Mode II: (**a**) in-plane stress contour and (**b**) crack growth length at applied displacement.

In Figure 16a,b, which show the displacement–load curve as the analysis result of the Mixed-mode and the crack growth behavior and fracture toughness, respectively, it can be observed that the test and analysis exhibited similar trends. In the test, a 3 mm pre-crack

test was performed to ensure the homogeneity of the initial crack shape. Additionally, the analysis considered that crack growth behavior was confirmed from 25 mm (including pre-crack length) because crack propagation was generally simulated as a cohesive failure.

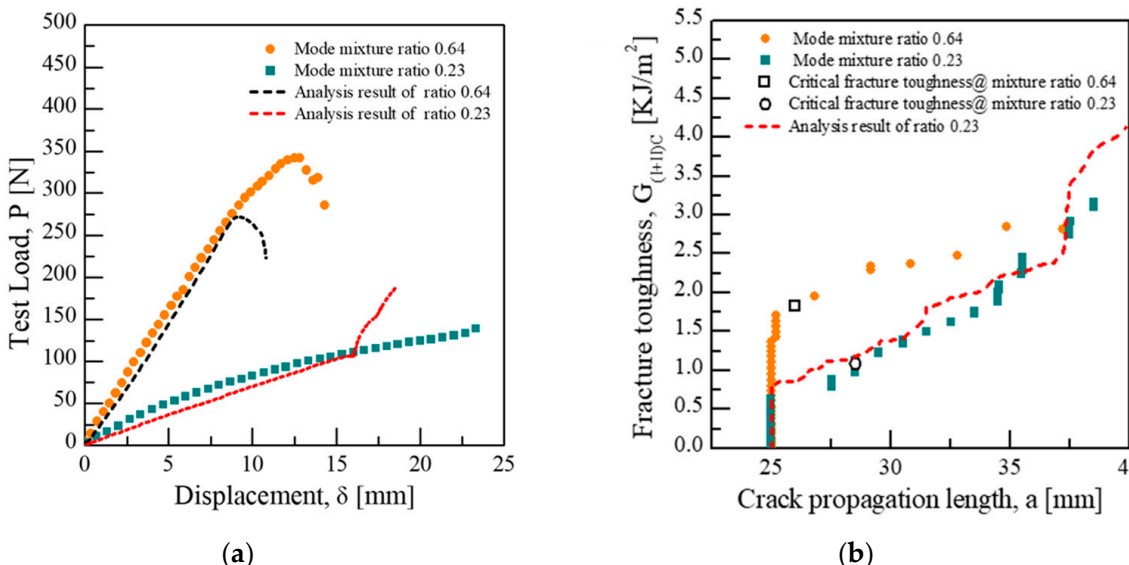

(**a**)                                                                              (**b**)

**Figure 16.** Comparative analysis of specimen-level FE analysis results and Mixed-mode test results: (**a**) $P - \delta$ curve and (**b**) $G_{(I+II)} - a$ curve.

On the other hand, in Figure 17, crack growth was observed as the stress propagated from the center of the specimen to the edge. Figure 17a shows the initial crack selection, the shape of the final crack, and the in-plane stress as the principal stress, and Figure 17b shows the crack growth behavior with respect to the applied displacement and the average stress at the critical point. Particularly, Figure 17b shows a similar trend as the fracture surface shown in Figure 11a based on the locations of the initial crack and crack growth. In other words, the reliability of the crack growth model in the Mixed-mode was established by confirming the similarity between the progressive fracture analysis method implemented through a finite element analysis and the test. Therefore, to reduce the deviation of the test and analysis results at the initial crack location, lamination patterns and surface treatment with a relatively smaller bridging effect in the test are required. Consequently, it is believed that the characteristics of the composite-adhesive-joint failure must be considered when evaluating the durability of the blades for a wind turbine.

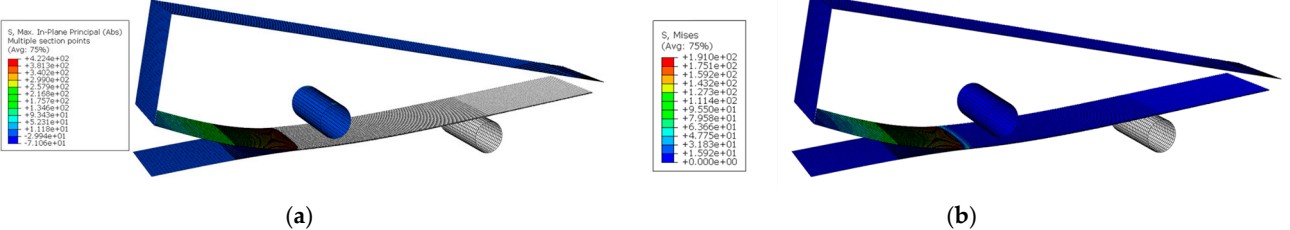

(**a**)                                                                              (**b**)

**Figure 17.** Fracture cross-section of Mixed-mode analysis results: (**a**) in-plane stress contour and (**b**) von Mises contour.

### 3.3. Progressive Failure Analysis for Full-Scale Composite Blade

The analysis was performed by applying the CZM method that was verified in Section 3.2. The initial failure of the composite blade joint begins at 86% of the reference load, and the stress generated in the spar cap–shear web adhesive joint increases rapidly and reaches the cohesive strength of 68 MPa, causing initial failure. The location of the initial failure is approximately 15 m from the spar cap–shear web, and the predicted location of

the failure in the TE area is near the root, as shown in Figure 18. In addition, a progressive failure of the adhesive joint occurs at 90% of the load scale. Subsequently, an additional failure in the spar cap–shear web occurs on the pressure side (PS), and a stress of 44 MPa is simultaneously generated in the TE area on the suction side (SS), confirming the increased likelihood of progressive failure.

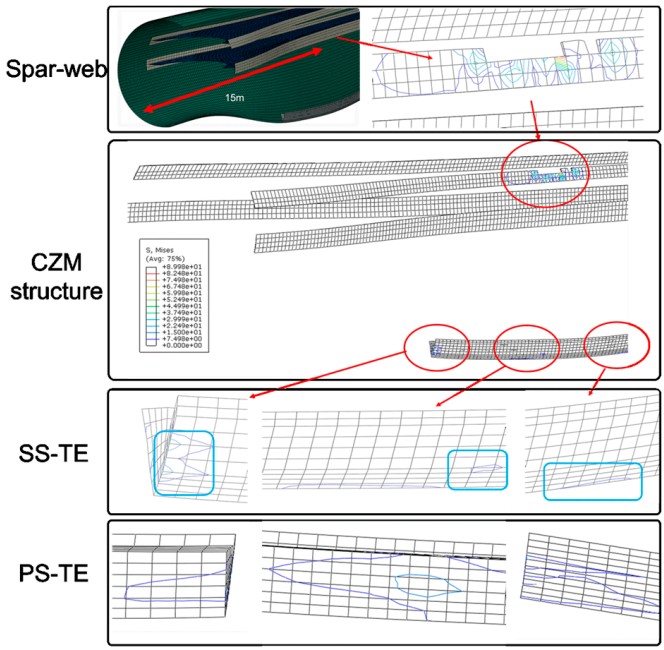

**Figure 18.** Crack growth shape depending on the wind blade scale: 86% load scale.

Finally, the analysis was performed up to 200% of the load scale, and a progressive failure was observed in the direction of the root from the initial failure location of 15 m, as shown in Figure 19. In the TE area, a progressive failure occurred at 160% of the load scale after the initial failure, which was at 140% of the load scale.

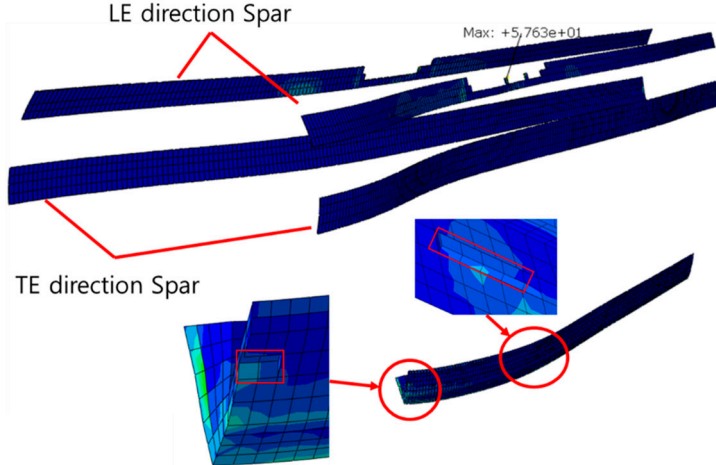

**Figure 19.** Crack growth shape depending on the wind blade scale: 200% of the load scale.

While the durability of this model is ensured at 100% of the load scale for a general structural analysis, the adhesive failure in the joint occurred at 86% before the reference load, and the failure was progressed. As a result, the failure deteriorated progressively in the spar cap–shear web adhesive joint near the LE, approximately 12–15 m from the root. Consequently, the behavior of the blade became unstable, and the risk of local buckling was

observed to have increased, as shown in Figure 20. In addition, as the failure in the spar cap–shear web adhesive joint deteriorated, the principal stress generated in the composite material rapidly increased. The tensile stress was 746.8 MPa, and the compressive stress was 437.2 MPa, confirming the increased failure likelihood of the composite material due to the tensile stress.

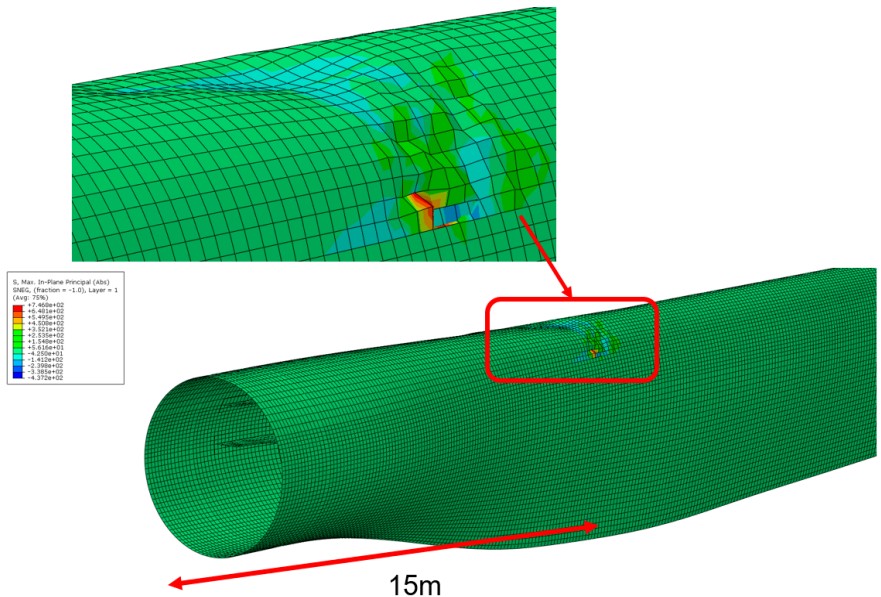

**Figure 20.** Crack growth shape depending on the wind blade scale and buckling risk: 200% of load scale.

As shown in Figure 21, the quadratic nominal stress (QUADS damage) standard was used to examine the high probability of cracking of the adhesive. First, the QUADS failure factor at 90% of the reference load indicated that failure occurred progressively from the adhesive joint of the LE, while no initial failure was observed in the TE, as shown in Figure 21a. In addition, at 200% of the reference load, the TE exhibited a low failure factor, while additional failure was observed in the adhesive joint of the LE, as shown in Figure 21b. It was confirmed that these characteristics caused the initial adhesive failure of the joint to the blade behavior due to torsion according to the displacement contour across to the entire TE. Additionally, it was identified that failure after the initial behavior occurs progressive due to the flap direction load. Therefore, it was confirmed that additional consideration is required for predicting the likelihood of the initial failure in the adhesive joint due to torsion when designing wind turbine blades based on the failure tendency.

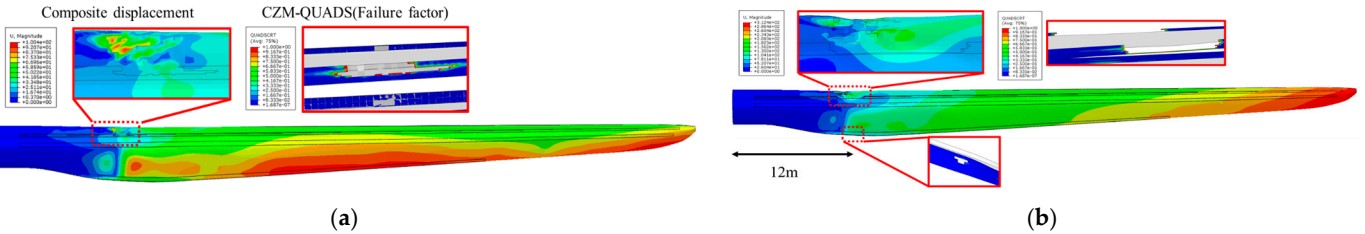

(**a**)　　　　　　　　　　　　　　　　　　　　　　　　　　　　　　　　　　(**b**)

**Figure 21.** Displacement and failure shape depending on the full-scale blade: (**a**) 90% of reference load and (**b**) 200% of reference load.

## 4. Conclusions

Currently, the designers for composite wind turbine blades have used the first failure criteria criterion (maximum stress/strain, Tsai-Wu, Tsai-hill). However, although sufficient structural integrity is secured in terms of safety margin, various damages have been reported in real composite blades. Among them, it is reported that the debonding damage

in the spar cap–shear web joint is the most frequently occurring and dangerous damage mode. For this problem, many researchers have focused on the initiation mechanism, geometry, and locations of debonding damage. Furthermore, these will be used to evaluate the debonding damage propagation behavior under the operating conditions and fatigue load spectrum acting on the blade. Additionally, this paper deals with the debonding damage initiation mechanism, geometry, and initiation locations through the application of the progressive failure analysis method as follows.

In this study, to evaluate the debonding fracture characteristics of the hybrid composite laminated joint applied to the wind turbine composite blade, the crack initial/growth prediction model of the adhesive joint was proposed using the fracture toughness test, fracture model verification, and progressive failure analysis. Using this, the failure mechanism of debonding damage on adhesive joints of MW-class wind turbine blades was analyzed. In particular, the damage started from 86% of the ultimate load applied to the NREL 5MW-class blade, and then local buckling occurred. Therefore, the risk of blade breakage was predicted during the operation stage during the design life span of 25 years, and it was reviewed that evaluation considering joint debonding damages is necessary for high-reliability design along with the existing blade safety evaluation method [20].

**Author Contributions:** Formal analysis, H.K.; Resources, Y.J.; Project administration, K.K. All authors have read and agreed to the published version of the manuscript.

**Funding:** This work was partly supported by the Korea Institute of Energy Technology Evaluation and Planning (KETEP) grant funded by the Korea government (MOTIE) (No. 20213030020120) and the Korea Institute of Energy Technology Evaluation and Planning (KETEP) grant funded by the Korea government (MOTIE) (20224000000220, Jeonbuk Regional Energy Cluster Training of human resources).

**Institutional Review Board Statement:** Not applicable.

**Informed Consent Statement:** Not applicable.

**Data Availability Statement:** Not applicable.

**Conflicts of Interest:** The authors declare no conflict of interest.

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
