# Peer review of "Progressive Failure Analysis for 5MW-Class Wind Turbine Composite Blades with Debonding Damage based on CZM Method"

_applsci, doi:10.3390/app122412973_

Round 1

Reviewer 1 Report

This paper presents a CZM method to identify the failure mechanism in the adhesive joints between the spar cap – shear web and trailing edge of composite blades. It calculated fracture toughness through Mode I, Mode II, and Mixed-mode tests for quantitative analysis of adhesive joints. It calculated fracture toughness through Mode I, Mode II, and Mixed-mode tests for quantitative analysis of adhesive joints. Overall, it is an interesting topic and the topic fits the journal very well. However,this paper can be further improved if the following questions are answered/discussed.

1. What is the advantages of CZM method compare to other methods?

2. What does Eq. (2) mean? What is the relationship between this equation and the following parts of the paper? It will be better if it can be indicated clearer.

3. It does not need “space” before where.

4. When CZM firstly appeared, its full name should be indicated, i.e. Cohesive Zone Modeling (CZM);

5. It will be much clearer if CZM method and how it applied to analyze Wind Turbine with debonding damage can be presented more.

6. For now, the shape of specimen is a thin plate, but the wind blade has its own airfoil. Does the shape of wind blade affect your results?

Author Response

1. 

  • Virtual Crack Closure Technique (VCCT) and Cohesive Zone Modeling (CZM) techniques are typically used for analysis to implement blade cohesive , and CZM has the advantage of being easy to express thickness and nonlinear behavior.

2.

  • Eq.(2) represents the fracture growth critierion in the theoretical method for joint damage simulation, the formula is a theoretical method for realizing the growth behavior after initial failure. Also, the expression of Eq. (2) has been corrected/supplemented.

3.

  • Corrected blanks in front of “where” in lines 86, 212, and 238.

4.

  • Added abbreviation of CZM (Cohesive Zone Modeling) in the abstract.

5.

  • In order to analyze the damage, a lot of information such as blade airfoil and stacking information is required, and it must be a reliable model. Therefore, in this study, the NREL(National Renewable Energy Laboratory) model, which is a reference model used in many studies, was selected. In addition, in order to increase the reliability of the method mentioned in this paper, we will conduct a study on the damage using CZM analysis for the models to be developed in the future.

6.

  • The spar cap-shear web and trailing edges, which have significant debonding characteristics in large blades, are not areas with many curvature variations. In addition, since the adhesive properties were acquired quantitatively based on the ASTM standard, we proceeded with the judgment that there was no effect on curvature.

Reviewer 2 Report

It is a good research written by the author where the author clearly explain on progressive failure analysis for 5MW class wind turbine 2 composite blade with debonding damage based on CZM 3 method. However, in the abstract, author should summarize clearly on the result of this study by explaining how the method CZM3 applied totally give impact to the wind turbine. Conclusion and recommendation also should be provided in the abstract. 

On the reference part, I can say that some of references are too outdated since there is a reference is from 1989, and 1992. Author should find references 5 years recent in order to support the discussion part of this study. 

The rest of this paper can be considered as a good research since author applied right methods and supported result of this study with suitable graphs and figures that in line to every objectives of this study. 

Author Response

1.

  • The 1989 and 1992 references are for pioneers who developed and shared the theory used in this study. The latest Information on related research is presented as a trend in the introduction.

2.

  • The conclusion was revised and added from “In this study, to evaluate the debonding fracture characteristics of the hybrid compo-site laminated joint applied to the wind turbine composite blade, the crack initial/growth prediction model of the adhesive joint was proposed using the fracture toughness test, fracture model verification, and progressive failure analysis. Using this, the failure mech-anism of debonding damage on adhesive joints of MW-class wind turbine blade was analyzed.”
     to “In this study, to evaluate the debonding fracture characteristics of the hybrid composite laminated joint applied to the wind turbine composite blade, the crack initial/growth prediction model of the adhesive joint was proposed using the fracture toughness test, fracture model verification, and progressive failure analysis. Using this, the failure mechanism of debonding damage on adhesive joints of MW class wind turbine blade was analyzed. In particular, the defect started from 86% of the ultimate load applied to the NREL 5MW class blade, and then local buckling occurred. Therefore, the risk of blade breakage was predicted during the operation stage during the design life span of 20 years, and it was reviewed that evaluation considering joint debonding defects is necessary for high reliability design along with the existing blade safety evaluation method [20].”.

Reviewer 3 Report

This paper deals with the Progressive Failure Analysis for 5MW Class Wind Turbine Composite Blade with Debonding Damage based on CZM method. While the experimental and numerical aspects of the problem seems to be interesting; unfortunately, nothing contribution is observed in the analytical part of the issue. The pros and cons of this study with respect to other methods implemented in progressive failure analysis can be hardly figured out. Moreover, the conclusion section is provided in a very briefly manner.

Author Response

The conclusion was revised and added from “In this study, to evaluate the debonding fracture characteristics of the hybrid compo-site laminated joint applied to the wind turbine composite blade, the crack initial/growth prediction model of the adhesive joint was proposed using the fracture toughness test, fracture model verification, and progressive failure analysis. Using this, the failure mech-anism of debonding damage on adhesive joints of MW-class wind turbine blade was analyzed.”

to “In this study, to evaluate the debonding fracture characteristics of the hybrid composite laminated joint applied to the wind turbine composite blade, the crack initial/growth prediction model of the adhesive joint was proposed using the fracture toughness test, fracture model verification, and progressive failure analysis. Using this, the failure mechanism of debonding damage on adhesive joints of MW class wind turbine blade was analyzed.

In particular, the defect started from 86% of the ultimate load applied to the NREL 5MW class blade, and then local buckling occurred. Therefore, the risk of blade breakage was predicted during the operation stage during the design life span of 20 years, and it was reviewed that evaluation considering joint debonding defects is necessary for high reliability design along with the existing blade safety evaluation method [20].”.

Reviewer 4 Report

The only note is about the conclusion.

After all this long and interest work, you need to have a very scientific and interest conclusions and recommendations. 

I really want to see these two sections in the next version. 

Author Response

(The authors gave the same response as above.)

Round 2

Reviewer 3 Report

Previous concerns still exist.

Author Response

The author understands that your comment indicates that this paper has no contribution to the analytical field of progressive failure analysis. In other words, we authors think that this paper have no theoretical contributions to the field of progressive failure analysis, although the experimental or numerical methodology is interesting. As pointed out, this paper does not aim to improve the analytical or theoretical parts of the field of progressive failure analysis, but rather to improve the design methodology of composite wind turbine blades by introducing progressive failure analysis techniques. there is. Please, let us explain this in more detail below.

Currently, design for composite wind turbine blades do not consider the failure of structural integrity of constituents of composite materials (fiber reinforcements, matrix systems) or very complicated failure modes (matrix cracking, delamination, debonding or fiber breakage) that may occur, and simply the first failure criterion (maximum stress/strain, Tsai-Wu, Tsai-Hill) and safety factor are simply applied. However, although sufficient structural integrity secured in terms of design, various damages have been reported in real composite blades. In particular, many debonding damage have been reported in the spar cap-shear web joint where the suction and pressure sides of the blade should be bonded. 

Currently, the composite wind turbine blade research group has focused on the debonding occurrence condition, its mechanism, and its growth (steady state state) of these spar cap-shear web joints. The authors, as the members of the research group, have also been conducting the research over the past several years, and have identified the mechanism, location and size of debonding damage through the application of the progressive failure analysis method. We are also planning to conduct a study on how debonding damage propagates under the operating load (fatigue load) acting on the blade. and I think that the research for the debonding damage initiation mechanism, geometry, and initiation locations through the application of the progressive failure analysis technique of this paper is one of the most important research for innovation on design of composite wind turbine blade. 

According to the reviewer's comment, the above contents were implemented in the conclusion in this manuscript to accurately explain the academic significance of this manuscript. 
